# Utility of carbon and nitrogen stable isotopes for inferring wild bee (Hymenoptera: Apoidea) use of adjacent foraging habitats

**Jessie Lanterman Novotny**[1,2]*, **Karen Goodell**[2]

**1** Department of Biology, Hiram College, Hiram, Ohio, United States of America, **2** Department of Evolution, Ecology, and Organismal Biology, The Ohio State University, Newark, Ohio, United States of America

* JessieLantermanNovotny@gmail.com

**Data Availability Statement:** All relevant data are within the paper and its Supporting information files.

**Funding:** This research was funded by the Ohio State University Graduate School's Alumni Grants

## Abstract

Isotope analysis has proven useful for understanding diets of animals that are difficult to track for extended periods. Bees are small yet highly mobile and often forage from multiple habitats. However, current methods of assessing diet are limited in scope. Efficient methods of tracking bee diets that integrate across life stages, distinguish habitat use, and are sensitive to taxonomic differences will inform conservation strategies. We evaluated the utility of stable isotope analysis for estimating contributions of adjacent habitats to bees' diets. We also investigated taxonomic variation in bee and flower isotope composition. We measured natural abundance of carbon and nitrogen stable isotopes in two body regions from three wild bee genera, as well as in 25 species of flowers that likely comprised their diets. Bee $\partial^{13}$C and $\partial^{15}$N varied with habitat and taxonomic groups (conflated with month), but did not match spatial or seasonal trends in their food plants. Flower $\partial^{13}$C was lowest in the forest and in April–June, as expected if driven by water availability. However, bee $\partial^{13}$C was elevated in the spring, likely from overwintering nutritional stress or unpredictable food availability. Bumble bees (*Bombus*) were enriched in $\partial^{15}$N compared to others, possibly reflecting differences in larval feeding. Bee diet mixing models had high variation and should be interpreted with caution. Models estimated similar habitat contributions to diets of spring *Andrena* and overwintered *Bombus* queens. Summer *Bombus* queens and workers were indistinguishable. Sweat bees (*Halictus*) were estimated to use comparatively more field flowers than others. Overall, taxon more strongly influenced isotope composition than either foraging habitat or month, likely because of associated differences in sociality and timing of annual activity. Future studies seeking to reveal bee diets by isotope analysis may gain better resolution in more isotopically distinct habitats, in conjunction with controlled feeding or isotope labeling experiments.

## Introduction

Bees are widely considered the most efficient insect pollinators, but they are under threat from multiple anthropogenic stressors [1] including habitat loss and fragmentation [2]. Habitat

for Graduate Research and Scholarship (AGGRS) Program.

**Competing interests:** The authors have declared that no competing interests exist.

conservation and wildflower plantings are the principal management interventions for bee conservation, but our understanding of how bees use these enhanced habitats (typically grasslands) and move between adjacent habitats, such as forests, edges, wetlands and developed areas over the season is incomplete [3, 4]. Many bee species use resources from multiple habitats to meet their lifelong nesting and nutritional needs [e.g., 5]. Understanding the extent to which bees forage from different habitats within a landscape over a lifetime will help inform habitat conservation and management decisions [6].

Traditional methods of monitoring bee diets and habitat use (such as netting and observing bees foraging on flowers) are costly in terms of researcher time and limited by experience. They are also highly sensitive to flower availability and weather conditions. Field observation data are incomplete as they provide only snapshot measures of bee resource use at a certain place and time, not across the entire range of their daily or annual activity. Thus, a large portion of bees' diets may go unrecorded using traditional monitoring methods, particularly in areas that are inaccessible without special equipment, like tree canopies or dense vegetation. Identification of pollen from foragers and larval nest provisions offers additional insight into bee diets over a larger time scale, and potentially the landscape scale, if pollens can be assigned to plant species unique to particular habitat types with a comprehensive and localized pollen reference collection [7–9]. Morphological identification of pollen requires significant expertise and is time-consuming, although recent developments in DNA-based techniques, such as pollen meta-barcoding show promise for increasing efficiency of pollen identification [10]. Nevertheless, larval pollen analysis depends on researchers' ability to locate and excavate nests.

Stable isotope analysis of bee and flower tissue has the potential to solve these challenges in understanding bee diets by offering an integrated measure of bees' habitat use across their life span and revealing differences among taxa [11]. However, most previous bee studies using stable isotopes have focused on economic rather than ecological questions [12–14]. Isotopes have been seldom used to discriminate bee foraging patterns [11, 15]. Brosi et al. [11] observed differences in $\partial^{13}C$ of bees foraging exclusively in either forest or pasture, and between wet season versus dry season bees, but they did not test the isotopic composition of floral resources and therefore were unable to resolve what proportion of an individual's diet came from one habitat versus another for bees foraging across multiple habitats. In our study we seek to fill this gap by isotopically inferring the relative importance of adjacent foraging habitats for wild bees, based on the isotope signature of flowers that likely comprised their diets in each habitat.

Carbon and nitrogen stable isotopes are often used to trace the flow of nutrients through food webs and across ecosystems [16–18]. Both elements have naturally-occurring stable isotopes, of which the heavy versions ($^{13}C$, $^{15}N$) make up less than 1% of their natural abundances [19]. Discrimination against the heavier isotopes (*e.g.* $^{13}C$ versus $^{12}C$, $^{15}N$ versus $^{14}N$) in biochemical reactions, or isotope fractionation, is measured as the ratio of the heavy to light isotope in a sample relative to a known standard (expressed as $\partial^{13}C$ ‰, $\partial^{15}N$ ‰). Isotope analysis has proven useful for understanding diets of other animals that are hard to track [*e.g.* 20, 21]. Bees stable isotope composition may reveal integrated information about their lifelong diets that traditional monitoring methods miss, as many species have broad foraging ranges (up to several kilometers from their nest) that encompass multiple habitats [22]. However, it requires that bees' food plants be isotopically distinct in space and time.

Plants vary in isotope signature with habitat and time of year due to water availability. Physiologically stressful growing conditions leave their mark isotopically on plants, and thus on the animals that feed on them [23]. During photosynthesis, more $^{13}C$ is incorporated into plant tissues in times of water stress when stomata are closed and $CO_2$ is not being refreshed [16]. Therefore, over the growing season as water becomes scarce, plants incorporate more $\partial^{13}C$ [24]. Similarly, when soil nitrogen is limiting, plants should incorporate more of the

heavy $^{15}$N isotope [17]. Nitrogen is presumed to be more limiting in non-agricultural grasslands (hereafter, fields) than in forest for two reasons: (1) because forests have enhanced nitrogen recycling pathways that increase N availability to plants [25, 26]; and (2) volatilization of ammonia from soil concentrates the heavy isotope in the drier more exposed field soils [27, 28]. Therefore, plants growing in the field should be enriched in the heavy C and N isotopes compared to those growing in moist, nutrient rich forest.

Because of bioaccumulation we expect stable isotope ratios of bees to reflect spatial and seasonal patterns in their food plants. Specifically, bees feeding on forest flowers in the spring should be lighter, or depleted, in $\partial^{13}$C and $\partial^{15}$N than bees feeding primarily on field flowers later in the season [11]. Bees' choice of foraging habitat is driven by changing availability of flowers across the season [5, 8, 22]. In southeastern Ohio, where this study was conducted, the landscape is dominated by hardwood forest interspersed with pockets of meadow and agricultural lands. In spring, forests and forest-field edges offer plentiful understory herbs, shrubs, and trees as forage for bees. In mid- to late summer, fields offer the most abundant flowers [22]. Bee taxa that emerge at different times of the season collect pollen and nectar from different plant species, or from plants of the same species grown under differing water and nutrients conditions.

Differences in bee isotope signatures by habitat and season may be underpinned by differences among taxa in the timing of annual activity, sociality, and body size. Most bee species in the study region are solitary or semi-social with adults that are active for one to two months out of the year beginning in late spring or early summer [29]. Two common taxa exemplify this variability: the solitary spring mining bees (*Andrena* spp.) and primitively social summer sweat bees (*Halictus* spp.). Bees of both genera are small to medium sized (5-15mm in length) and forage mostly within 1000m of their nest [30, 31]. These taxa are likely to be more restricted in habitat use than larger bees that have broader foraging ranges. Bumble bees (*Bombus* spp.), in contrast, live in annual eusocial colonies that are active for most of the summer. Bumble bee colony longevity, high colony resource requirements, and generalized diets make them more likely to use flowers from multiple habitats.

The aim of this study was to evaluate the usefulness of stable isotopes for tracking habitat use by foraging wild bees across the growing season. If bees' isotope signatures reliably reflect their primary feeding grounds, we can use these data to identify key habitats to manage for bee conservation. We tested the following predictions:

1. Flower $\partial^{13}$C and $\partial^{15}$N will differ by habitat and month. Specifically, flowers grown in conditions that are presumed to have higher water and N limitation (field habitat, and later in the season) should be elevated in $\partial^{13}$C and $\partial^{15}$N.

2. Bee $\partial^{13}$C and $\partial^{15}$N will reflect spatial and seasonal trends in the isotope composition of their food plants.

3. Bee $\partial^{13}$C and $\partial^{15}$N will vary with taxon due to differences in the timing of their annual activity and other life history traits.

## Materials and methods

### Study system

Once per month in April–August 2016, foraging bees of three common genera and flowers from their preferred food plants were collected from adjacent habitats (a grassland-reclaimed coal mine, adjacent remnant forest, and intervening forest edge). These specimens were analyzed for isotopic fractionation of $^{13}$C and $^{15}$N to infer spatial and temporal foraging patterns.

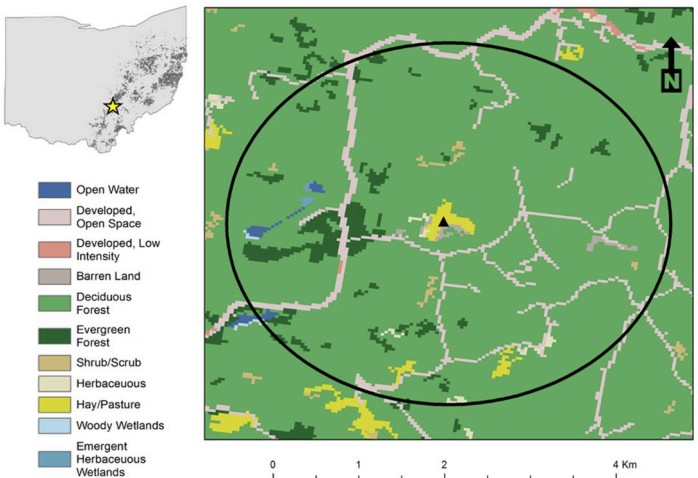

**Fig 1. Land use within a 2km buffer area of the study site in Wayne National Forest, New Straitsville, OH.**
Location in Ohio marked with a star on state map. The extent of current surface and underground coal mining
operations and abandoned mine lands in Ohio are shown in dark gray.

The study site was in Wayne National Forest (New Straightsville, Perry Co, Ohio,
39.55494˚N, -82.24743˚W). Special use authorization for research at this site was granted by
authorized officer Lynda Andrews, United States Department of Agriculture, Forest Service
Wildlife Biologist, in accordance with National Forest land use guidelines. It consisted of a
17.4 ha former surface coal mine that was reclaimed to grassland approximately 15 years ago
by the US Forest Service and was surrounded by remnant forest. Within a 2km buffer radius
of the site center, the landscape was 91.0% forested (deciduous—86.3%, evergreen—4.7%),
6.1% developed, 1.4% pasture/herbaceous habitat, and 1.5% of open water, barren unre-
claimed coal spoils, and other land use types (National Land Cover Data 2011 [32]; analyzed
in ArcGIS software [33], Fig 1). The landscape setting made the site ideal for this study
because it created two clearly delineated habitats that support bees at different times during
the season. Bees foraging in the grassland and adjacent forest were unlikely to be foraging
elsewhere because the scale of the surrounding forest exceeds the typical foraging radius of
most bees. Based on the literature, we also assumed that forest and field had differences in
water and nitrogen availability that translated into differences in isotope fractionation of the
plants [24–28].

This site is representative of the unique landscape in SE Ohio. Coal mining has been exten-
sive (Fig 1) and reclamation has produced tens of thousands of acres of newly-created grass-
land habitat (http://minerals.ohiodnr.gov), which may act as resource islands for pollinators in
disturbed landscapes [34]. Reclamation grasslands have shallow soils that do not freely revert
to forest, but remain open-grown and undeveloped for many years [35, 36]. Over time they
develop complex plant and pollinator communities, and therefore may provide foraging and
nesting resources needed to sustain wild bees [37].

## Bee field sampling

Three bee genera were chosen to represent the bee fauna of our study region–spring mining
bees (*Andrena* spp.), summer sweat bees (*Halictus* spp.), and bumble bees (*Bombus* spp.).
*Andrena* is a diverse genus of mostly small–medium sized solitary bees that are active in the
spring and early summer and believed to feed primarily on forest and edge flowers. Many

*Andrena* species are also known to specialize on certain flower taxa. *Halictus* are small-bodied summer bees with generalist diets that have very small eusocial colonies, and are known to nest and forage in fields. Bumble bees (*Bombus* spp.) are large-bodied diet generalists with long periods of seasonal activity at the colony level, and should change their foraging habitats across the season to track shifts in flower availability [38]. Across the season, we collected over-wintered *Bombus* queens (April-May), female *Andrena* (May), *Bombus* workers (May—Aug), female *Halictus* (June and July), and new *Bombus* queens-of-the-year (July-Aug).

On each sample date, we observed bees foraging for 1 hour and identified the most-visited food plants in an area that encompassed the reclamation grassland (hereafter "field"), the adjacent remnant forest (hereafter "forest"), and a 20 m wide transitional edge zone in between (hereafter "edge"). Bees were then netted into sterile glass vials from these target flower species for 4 hours. Field and forest surveys were conducted at least 20 m from the edge into the habitat type. Institutional Animal Care and Use Committee approval was not required for this research as no vertebrate animals or cephalopods were used in this study. Nonetheless, bee specimens were treated with care and euthanized by freezing to minimize suffering. To help account for the possibility of recent food lowering $\partial^{13}$C in bee abdomen, in the field bees were left in the vials for a minimum of two hours before freezing and they typically defecated and regurgitated nectar from the crop during that time before they were euthanized. We recorded foraging habitat for all netted bees, and identified them to species in the lab using taxonomic keys available from Discoverlife [39]. No bees were observed visiting flowers in the forest understory in 6 hours of netting in April and May (peak flowering time in that habitat).

## Flower field sampling

To assess the availability of bee forage across the season, once per month we surveyed flower diversity and abundance along two 100 m x 1 m transects, one in the field and one along the edge. We were unable to quantify forest flowers because no flowers were found in the forest understory in June–August. On each sample date, fresh flowers presenting pollen and/or nectar were collected from 3–5 individuals of each of three wildflower species visited by our target bee genera. Flowers were placed in sterile vials by species to represent likely dietary sources of bee C and N. Whole flowers were used to account for possible differences in isotope fractionation between tissue types (petals, pollen, nectar), while also representing all the tissues from that flower of nutritional value to bees.

## Laboratory sample preparation & isotope analysis

Bees were washed with DI water to remove pollen, mites, and soil. The legs and wings were removed, and the abdomen was separated from the head plus thorax (hereafter "head+thorax") for separate analysis. Bee abdomens consist of chitin (exoskeleton), fat stores, digestive tissue, and reproductive tissue. The head+thorax consists mainly of chitin and flight muscles. Fat and reproductive tissue are more metabolically active than muscle, nervous, and exoskeleton and reflect recent adult diet [40–42]. Therefore, abdominal tissue of adult bees collected at different times of the season should vary in isotope signature with the flowers they have been feeding on within the past week [14, 41]. The head+thorax tissue more likely reflects larval diet from one month to one year prior (depending on the species' life cycle). Solitary and primitively social bee larvae feed on a pollen and nectar mass provisioned in their sealed brood cell by their mother. Bumble bee larvae are fed continuously by workers in the nest. Adults of most bee species are short-lived and subsist mainly on nectar. It has long been assumed that there is little remodeling of muscle and cuticle tissue during adulthood, although there is some isotopic

evidence that bees incorporate carbon from their adult diet into abdominal cuticle or adjacent tissues [14, see also S1 File for unpublished data from pilot study].

The abdomen and head+thorax were placed separately in sterile foil packets that had been baked in a muffle furnace for 4 hours at 500°C. Specimens were dried for 48 hours at 50°C, weighed, and ground to a powder with a sterile agate mortar and pestle. Between samples, all tools were wiped clean with a tissue, rinsed *ad libidum* with DI water, then with 70% ethanol, and allowed to air dry on sterile foil. The powdered sample was placed in a sterilized glass vial and stored in a 4°C freezer. *Halictus* abdomens (total mass < 3 mg) were cut in half lengthwise and broken into small pieces with forceps before being placed in tin capsules, rather than ground in the mortar and pestle, to prevent tissue loss. Of the dried homogenized powder, a $1.1 \pm 0.1$ mg sample from each bee and a $3.3 \pm 0.1$ mg sample of each flower (consisting of petal tissue plus pollen and nectar) was packed in a 5 x 8 mm standard weight tin capsule using a microbalance. The recommended amount of tissue was determined using the sample weight calculator from the UC Davis stable isotope facility (http://stableisotopefacility.ucdavis.edu/sample-weight-calculator.html), based on C and N content of bee (~10% N, 45% C) and flower (~2.5% N, 40% C) tissue in a pilot study (Burkhart and Goodell, *undergraduate thesis available from* https://kb.osu.edu/bitstream/handle/1811/80637/1/RobertBurkhartThesis.pdf). Encapsulated samples were analyzed at the University of California Davis Stable Isotope Facility for carbon and nitrogen isotope composition using an elemental analyzer coupled with an isotope ratio mass spectrometer under continuous flow [43]. The ratio of $^{13}C$:$^{12}C$ was calculated in parts per thousand (‰) deviation from Vienna PeeDee Belemnite limestone standard and denoted as $\partial^{13}C$. The ratio of $^{15}N$:$^{14}N$ was calculated in parts per thousand (‰) relative to atmospheric $N_2$ gas and denoted as $\partial^{15}N$. Duplicates were run for 10% of the flower and bee samples. Because biological tissues are depleted in $^{13}C$ compared to the international standard (Vienna PeeDee belemnite limestone), more negative $\partial^{13}C$ values are considered "lighter" or "depleted" in $^{13}C$. Similarly, a negative $\partial^{15}N$ value indicates that there is less of the $^{15}N$ than in the standard atmospheric $N_2$ gas. A positive value indicates that the sample is enriched in the heavy isotope relative to the standard.

To verify that age of individual bees did not affect isotope composition, we assessed wing wear as a proxy for bee age. The condition of the distal edges of each bee's forewings were scored on a 1–4 scale, modified from [44], by two independent observers: 1 = no damage (intact), 2 = minor indentations / damage on the wing margin (minor), 3 = most of the wing margin damaged (moderate), 4 = more than 5% of the wing missing (severe). The average of the two scores was then used in analyses.

## Data analysis

All analyses were performed in R version 3.3.1 [45]. The raw data are available in the Supporting Information as a.csv spreadsheet (S1 Table).

Isotope values of samples run in duplicate did not differ between the two measurements based on paired t-tests ($\partial^{13}C$ $t = 0.05$, $df = 16$, $p = 0.96$; $\partial^{15}N$ $t = 0.06$, $df = 16$, $p = 0.95$). Duplicate samples were not included in analyses.

We calculated lipid content of abdomen and head+thorax tissues using the equations from McConnaughey and McRoy [46]. Lipid content averaged $35.44 \pm 11.86$ in abdomen samples and $17.78 \pm 3.06$ in head+thorax tissues. Therefore, both abdomen and head+thorax $\partial^{13}C$ values were lipid normalized as in McConnaughey and McRoy [46]. Lipid normalization changed the abdomen $\partial^{13}C$ values by $1.30 \pm 0.75$ ‰ absolute value, but affected the HT samples negligibly ($0.19 \pm 0.16$ ‰) (**S2 Table in** S1 File).

Multiple linear models were used to investigate how habitat (edge, field), genus, and social caste (*Andrena* female, *Halictus* female, overwintered *Bombus* spring queens, non-overwintered *Bombus* summer queens-of-the-year, and *Bombus* workers) influenced bee abdomen $\partial^{13}$C and $\partial^{15}$N. Month was nested within genus-caste groups and was not included in the final model. Instead, seasonal trends were analyzed using a subset of data from one genus-caste group (*Bombus* workers) that was represented in all months. In univariate analyses, there was no effect of bee age (average wing wear score), the number of flowering species, or flower abundance on bee isotope signatures. Therefore, these variables were omitted from final models. There was not enough replication of bees on flower species to include the flower species on which a bee was collected as a random effect. Interaction terms were not included because not all group combinations were represented in the data.

Multiple linear models were used to investigate how habitat (edge, field, forest), month (Apr/May, June, July, and August), and plant growth form (woody, herbaceous) influenced flower $\partial^{13}$C and $\partial^{15}$N. We included growth form because univariate analyses showed that it had a larger effect on flower isotope composition than plant taxon (species, genus, or family level) or lifespan (annual, biennial, perennial), and others have analyzed isotope composition separately for woody versus herbaceous plants [47].

We further investigated the influence of all significant predictors on isotope composition using Tukey-adjusted least-squares means pairwise comparisons between levels within each variable (function *lsmeans*, package 'lsmeans'). To determine the overall effect of each predictor, model ANOVAs were computed using the Type III sum of squares method (simultaneous, rather than sequential). Variance inflation factors (VIFs) were calculated to check for multicollinearity among predictors (function *vif*, package 'car').

The proportion of individuals' diets that came from each habitat was estimated using isotope mixing models (MixSIAR, "long" model run with chain length = 300,000, burn in = 200,000, thin = 100, and chains = 3 [48]). Flowers collected from the field, forest, and edge habitats were treated as potential food sources, and the model attributed some proportion of each bee's C and N to each habitat, based on the proximity of the bee's isotope ratios to those of the food sources (considering variability in flowers within habitat) in the calculated "isotope mixing space" [49]. Although we did not observe any bees foraging in the forest, we know from previous experience and from the literature that many bees rely on spring wildflowers, shrubs, and trees in the forest. Also, one of the motivations of this study was to explore the use of isotope analysis for supplementing traditional ways of monitoring bee habitat use, particularly in habitats where observation of foraging bees is restricted by equipment, vegetation, and weather. Therefore, forest was retained in the model as a potential habitat food source. To the best of our knowledge there are no controlled feeding experiments with bees available in the literature, from which to calculate isotope fractionation between flower and bee tissue (but see S1 File text for results of a controlled feeding pilot study conducted in the Goodell lab). Therefore, trophic discrimination factors used in mixing models were calculated as the mean pairwise difference in $\partial^{13}$C$_{lipid\ normalized}$ and $\partial^{15}$N between bees and the flowers on which they were collected [50], with discrimination error as the standard deviation of the mean difference. During the mixing model run, the appropriate discrimination factors were added to flower isotope values to account for trophic offset between bees and their food plants.

## Results

In total, C and N isotope composition was measured in 89 female bees of 7 species (Table 1), and 25 species of flowering plants that were most likely to support bees at the site (Table 2). Bee abdomen and head+thorax isotope values showed similar trends by habitat, month, and

**Table 1. Bees sampled by species, social caste, and date.**

| Bee species | Caste | Habitat(s) | 27 Apr | 12 May | 28 Jun | 22 Jul | 20 Aug | Total |
|---|---|---|---|---|---|---|---|---|
| *Andrena carlini* | female | edge | | 16 | | | | 16 |
| *Andrena crataegi* | female | edge | | | 2 | | | 2 |
| *Bombus bimaculatus* | queen | edge, field | 3 | | | 2 | | 5 |
| *Bombus impatiens* | queen | field | 2 | | | | | 2 |
| *Bombus griseocollis* | queen | field | | | | | 5 | 5 |
| *Bombus bimaculatus* | worker | edge | | | 3 | 3 | | 6 |
| *Bombus impatiens* | worker | edge, field | | | 10 | 15 | 8 | 33 |
| *Bombus griseocollis* | worker | edge, field | | | 3 | 4 | 1 | 8 |
| *Halictus confusus* | female | edge | | | | 3 | | 3 |
| *Halictus ligatus* | female | field | | | 2 | 5 | 2 | 9 |
| **Total** | | | 5 | 16 | 20 | 32 | 17 | 89 |

genus-caste groups. The abdomen results are presented here because abdomen tissue was more likely to reflect recent adult diet (but see S1 File text for head+thorax analyses).

*Andrena* females each create a solitary nest. *Halictus* females may be workers or queens, although it is more likely our specimens were workers as queens remain in the nest after it is established. *Bombus* queens were separated from workers by size.

**Table 2. Flowers sampled by species and date.**

| Flower species | Family | Habitat(s) | 27 Apr | 12 May | 28 Jun | 22 Jul | 20 Aug | Total |
|---|---|---|---|---|---|---|---|---|
| *Achillea millefolium* | Asteraceae | Field | | | | X | | 1 |
| *Alliaria petiolata* | Brassicaceae | Forest | X | | | | | 1 |
| *Barbarea vulgaris* | Brassicaceae | Field | X | | | | | 1 |
| *Cirsium discolor* | Asteraceae | Field | | | | | X | 1 |
| *Cornus florida* | Cornaceae | Edge | X | | | | | 1 |
| *Daucus carota* | Apiaceae | Field | | | | X | | 1 |
| *Dipsacus fullonum* | Dipsacaceae | Edge | | | | X | | 1 |
| *Elaeagnus umbellata* | Elaeagnaceae | Field, Edge | X | | | | | 2 |
| *Geranium maculatum* | Geraniaceae | Forest | X | | | | | 1 |
| *Helianthus divaricatus* | Asteraceae | Field | | | X | X | | 2 |
| *Lamium purpureum* | Lamiaceae | Edge | X | | | | | 1 |
| *Lonicera morrowii* | Caprifoliaceae | Edge | X | X | | | | 2 |
| *Lotus corniculatus* | Fabaceae | Edge | | | X | X | X | 3 |
| *Melilotus alba* | Fabaceae | Edge | | | X | X | | 2 |
| *Monarda fistulosa* | Lamiaceae | Edge | | | | X | | 1 |
| *Packera* sp. | Asteraceae | Forest | X | | | | | 1 |
| *Phlox divaricata* | Polemoniaceae | Forest | X | | | | | 1 |
| *Ranunculus* sp. | Ranunculaceae | Forest | X | | | | | 1 |
| *Rubus pensylvanicus* | Rosaceae | Edge | X | | | | | 1 |
| *Rudbeckia hirta* | Asteraceae | Field | | | X | | | 1 |
| *Securigera varia* | Fabaceae | Edge | | | | X | | 1 |
| *Solidago canadensis* | Asteraceae | Field | | | | X | X | 2 |
| *Taraxacum officinale* | Asteraceae | Edge | X | | | | | 1 |
| *Trifolium pratense* | Fabaceae | Edge | | | X | | | 1 |
| *Trillium grandiflorum* | Liliaceae | Forest | X | | | | | 1 |

Total is the number of times that species was a dominant flowering resource for bees and was collected to represent a habitat for isotope diet end members.

## Bee isotope composition

In multiple linear models, bee $\partial^{13}C_{\text{lipid normalized}}$ and $\partial^{15}N$ were significantly influenced by genus-caste and by habitat (Table 3). Bees collected along the edge were enriched in $\partial^{13}C$ and in $\partial^{15}N$ relative to those caught in the field (Tukey-adjusted model pairwise comparisons $\partial^{13}C$ $t = 2.833$, df = 83, $p = 0.006$; $\partial^{15}N$ $t = 2.820$, df = 83, $p = 0.006$; Fig 2). Including genus-caste as a predictor explained isotope variation better than habitat alone, because it accounted for differences in timing of activity between spring miner bees (*Andrena* spp.) and summer sweat bees (*Halictus* spp.), as well as differences in life history and diet between bumble bee (*Bombus* spp.) queens and workers (Fig 3; pairwise model comparisons given in **S3 Table in** S1 File). Overwintered spring *Bombus* queens were significantly heavier in $\partial^{13}C$ and in $\partial^{15}N$ compared to *Andrena* and *Halictus*. Spring *Bombus* queens were also enriched in $\partial^{13}C$ compared to *Bombus* summer workers and queens-of-the-year. Summer *Bombus* queens-of-the-year were enriched in $\partial^{15}N$ compared to workers and somewhat enriched compared to overwintered spring queens. At the genus level, all *Bombus* were enriched in $\partial^{15}N$ compared to *Halictus* and *Andrena*. *Andrena* were not distinguishable from *Halictus* in $\partial^{13}C$ or $\partial^{15}N$. At the species level, two species (*Halictus ligatus* and *Andrena crataegi*) were significantly depleted in $\partial^{13}C$ ($F_{6,82} = 3.08$, $p = 0.01$), and two species (*H. ligatus* and *A. carlini*) were significantly depleted in $\partial^{15}N$ ($F_{6,82} = 40.107$, $p < 0.001$) compared to all others (Fig 3). Individuals showed little variability within a genus in abdomen $\partial^{13}C_{\text{lipid normalized}}$ (*Bombus* coefficient of variation CV = 2.52%, *Andrena* CV = 2.75%, and *Halictus* CV = 3.57%). However, *Andrena* and *Halictus* abdomen samples showed greater variability in $\partial^{15}N$ than those of bumble bees (*Andrena* CV = 1784.03%, *Halictus* CV = 221.37%, *Bombus* CV = 46.39%), possibly due to differences between taxa in brood provisioning versus larval feeding.

$\partial^{13}C$ values were lipid normalized following the methods of McConnaughey and McRoy (1979). Here we give the overall test statistic for each predictor and associated p values from model ANOVAs computed using Type III (simultaneous) sum of squares.

Seasonal trends were analyzed using a subset of data from one genuscaste group (*Bombus* workers) that was represented in all months, because in the full dataset month and genus were conflated due to inherent differences between taxa in the timing of their annual activity. *Bombus* workers were enriched in $\partial^{13}C_{\text{lipid normalized}}$ in June and July compared to August ($F_{2,44} = 9.628$, $p < 0.001$). Worker $\partial^{15}N$ was greater in July than in June and August ($F_{2,44} = 6.974$, $p = 0.002$) (Fig 4).

## Flower isotope composition

In univariate models, flowers growing in the forest were significantly lighter (more negative) in $\partial^{13}C$ than those growing in the field or along the edge ($F_{2,29} = 5.618$, $p = 0.009$; Fig 2). In multiple linear models, however, the effect of habitat on flower $\partial^{13}C$ was not significant. Instead, flower $\partial^{13}C$ was influenced by growth form (woody > herbaceous) and marginally by

**Table 3. The effects of habitat, genus, and social caste on wild bee abdomen C and N isotope composition.**

| Parameter | df | $\partial^{13}C_{\text{lipid normalized}}$ bee abdomen ($R^2$ adj. = 0.198) | | $\partial^{15}N$ bee abdomen ($R^2$ adj. = 0.520) | |
|---|---|---|---|---|---|
| | | F | p | F | p |
| Habitat | 1 | 8.028 | 0.006 | 7.952 | 0.006 |
| GenusCaste | 4 | 6.540 | <0.001 | 23.272 | <0.001 |

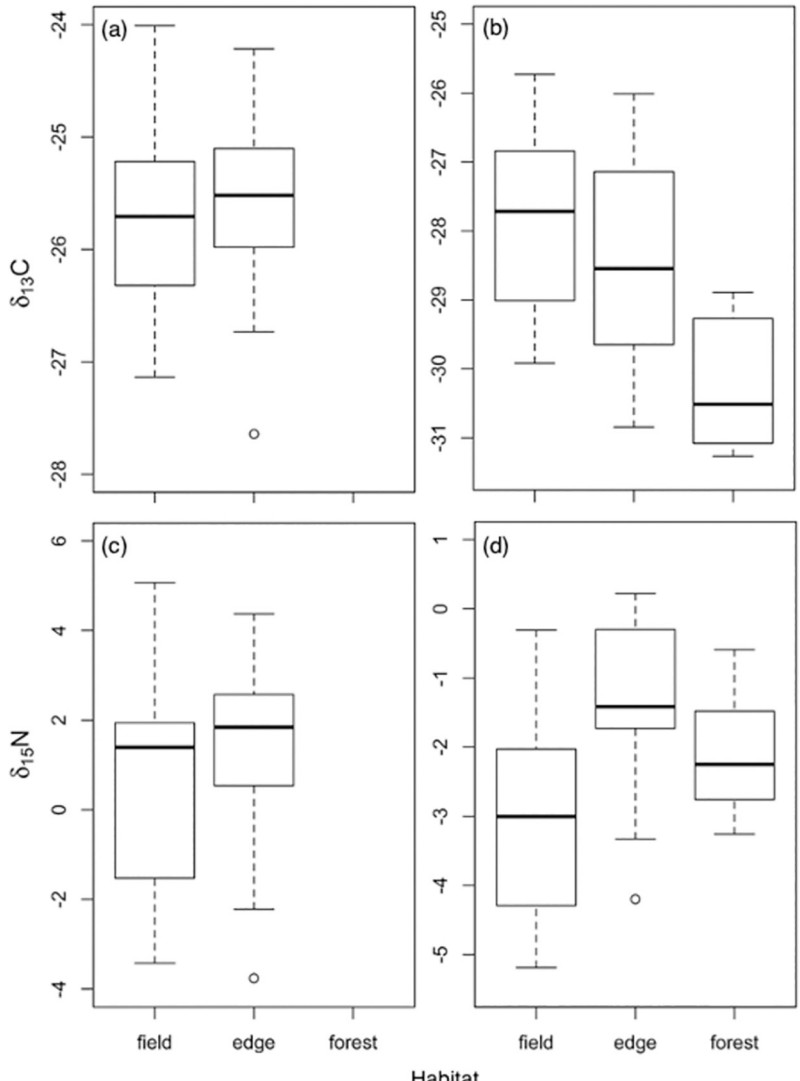

**Fig 2. Habitat trends in natural abundances of C and N stable isotopes in bee abdomens and flowers.** (a) bee $\partial^{13}C_{\text{lipid normalized}}$, (b) flower $\partial^{13}C$, (c) bee $\partial^{15}N$, (d) flower $\partial^{15}N$. The center line represents group median, with boxes showing upper and lower quartiles. The bars show value range. Bee sample sizes were as follows: field n = 25, edge n = 64, forest n = 0. Flower sample sizes were as follows: field n = 10, edge n = 16, forest n = 6.

month (Apr/May < July) (Table 4, see also **S4 Table in** S1 File for pairwise Tukey-adjusted model comparisons). It should be noted that because of changing bloom times of flowers across the landscape as the season progressed, habitat and month were somewhat conflated because the majority of flowers early in the season occurred in the forest and the majority late in the season were found in the field.

Models offered less insight into variation in flower $\partial^{15}N$, as the model $R^2$adj value was low ($R^2$adj = 0.201, Table 2). Instead, flower $\partial^{15}N$ varied significantly by species ($F_{25,6} = 3.469$, $p = 0.048$), and by family ($F_{25,6} = 6.449$, $p < 0.001$). Asteraceae had the lowest $\partial^{15}N$ values. Flowers in the Fabaceae family, many of which fix nitrogen via rhizobia and should therefore be depleted because they are less N-limited, were enriched instead in $\partial^{15}N$ ($F_{1,30} = 6.548$, $p = 0.016$).

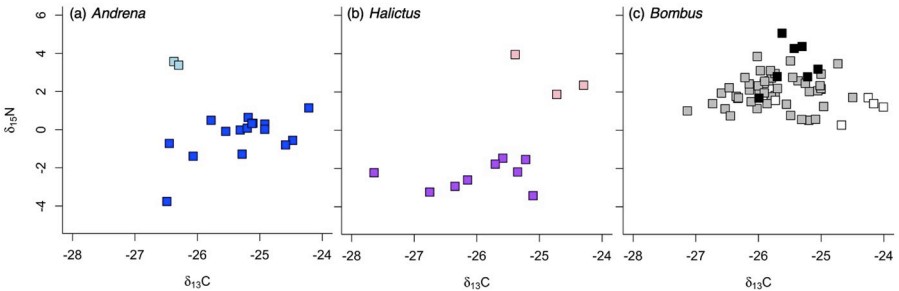

**Fig 3. Bee abdomen isotope composition by taxonomic and social caste groups.** (a) *Andrena carlini* (AprMay) are shown as dark blue squares, *A. crateaegi* (Jun) as light blue. (b) *Halictus ligatus* (Jun–Aug) are shown as dark purple squares, *H. confusus* (Jul) as light purple, (c) *Bombus* overwintered spring queens (AprMay) are shown as white squares, summer queens-of-the-year (Jul–Aug) as black, and summer workers (Jun–Aug) as gray. *Bombus* isotope values did not differ significantly by species. Bee $\partial^{13}C$ was lipid normalized following (McConnaughey and McRoy (1979).

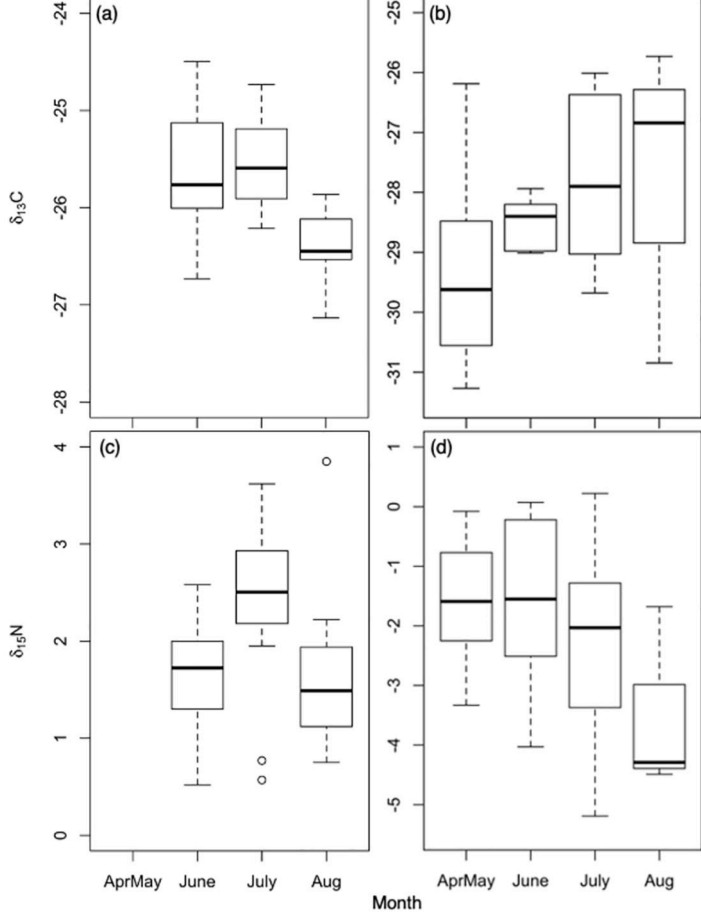

**Fig 4. Seasonal trends in natural abundances of C and N stable isotopes in *Bombus* worker abdomens and flowers.** (a) bee $\partial^{13}C_{\text{lipid normalized}}$, (b) flower $\partial^{13}C$, (c) bee $\partial^{15}N$, (d) flower $\partial^{15}N$. Samples collected at the end of April were combined with those from the beginning of May. The center line represents group median, with boxes showing upper and lower quartiles. The bars show value range. *Bombus* worker sample sizes were as follows: June n = 16, July n = 22, August n = 9. Flower sample sizes were as follows: AprMay n = 15, June n = 5, July n = 9, August n = 3.

**Table 4. The effects of habitat, month, and growth form on flower C and N isotope composition.**

| Parameter | df | $\partial^{13}$C flower (R$^2$ adj. = 0.370) | | $\partial^{15}$N flower (R$^2$ adj. = 0.201) | |
| --- | --- | --- | --- | --- | --- |
| | | F | p | F | p |
| Habitat | 2 | 0.785 | 0.467 | 3.404 | 0.049 |
| Month | 3 | 2.851 | 0.058 | 1.245 | 0.314 |
| Growth Form | 1 | 8.708 | 0.007 | 0.104 | 0.750 |

Here we give the overall test statistic for each predictor and associated p values from Model ANOVAs computed using Type III (simultaneous) sum of squares.

## Estimating habitat contribution to bee diets using isotope mixing models

There was high variation and overlap between potential food sources (Fig 5), so mixing model estimates of habitat contributions to bee diets should be interpreted with caution. Consistent with what is known about their foraging behavior, summer-flying sweat bees (*Halictus*) were estimated to rely more on field flowers than other genera (Table 5). *Bombus* workers and summer queens-of-the-year were indistinguishable in their diets, with the majority of their diets attributed to edge flowers. However, that is likely an indicator that the models did not resolve *Bombus* diets well because of their broad foraging ranges. Overwintered spring *Bombus* queens and *Andrena* were predicted to forage primarily on field and edge flowers rather than in the forest (Table 5), which is not consistent with other studies that have observed pollination of spring understory forest flowers.

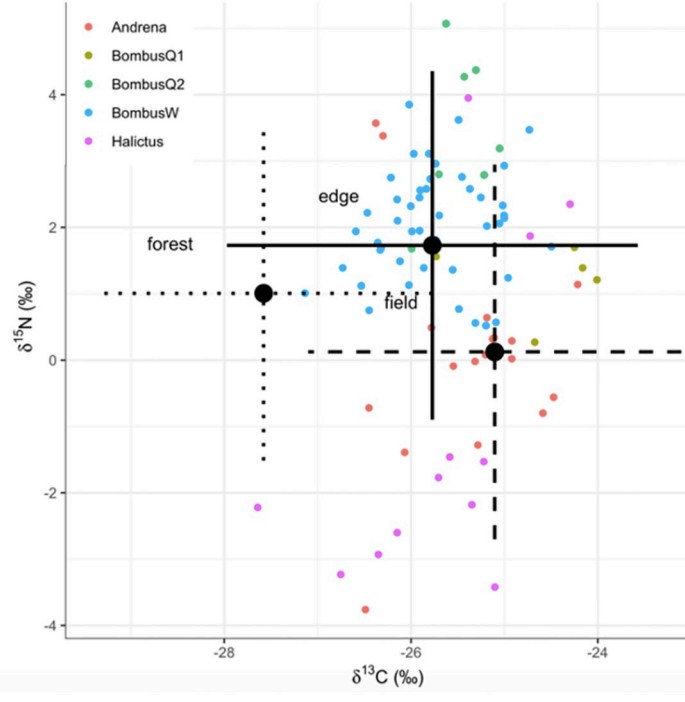

**Fig 5. Estimated contribution of field, forest, and edge flowers to bee diets.** Isospace mixing model plot generated using MixSIAR. Error bars indicate ± 1 SD (combined error for potential food sources and trophic discrimination factor).

**Table 5. Predicted proportion diet contributions of field, edge, and forest flowers to bee abdomen C and N isotope composition.**

| Bee Genus/Caste | Food Source | Mean Diet Contribution | Median | Lower | Upper |
|---|---|---|---|---|---|
| *Andrena* females | field flowers | 0.577 ± 0.121 | 0.579 | 0.334 | 0.796 |
| | edge flowers | 0.338 ± 0.111 | 0.336 | 0.120 | 0.558 |
| | forest flowers | 0.085 ± 0.078 | 0.066 | 0.003 | 0.288 |
| *Halictus* females | field flowers | 0.748 ± 0.154 | 0.764 | 0.416 | 0.969 |
| | edge flowers | 0.124 ± 0.089 | 0.103 | 0.014 | 0.342 |
| | forest flowers | 0.128 ± 0.138 | 0.085 | 0.001 | 0.459 |
| *Bombus* spring queens | field flowers | 0.656 ± 0.188 | 0.674 | 0.259 | 0.952 |
| | edge flowers | 0.296 ± 0.182 | 0.271 | 0.028 | 0.698 |
| | forest flowers | 0.048 ± 0.075 | 0.026 | 0.001 | 0.220 |
| *Bombus* workers | field flowers | 0.122 ± 0.079 | 0.108 | 0.019 | 0.313 |
| | edge flowers | 0.800 ± 0.151 | 0.832 | 0.333 | 0.963 |
| | forest flowers | 0.078 ± 0.110 | 0.041 | 0.001 | 0.414 |
| *Bombus* summer queens | field flowers | 0.111 ± 0.077 | 0.092 | 0.017 | 0.309 |
| | edge flowers | 0.825 ± 0.156 | 0.864 | 0.286 | 0.972 |
| | forest flowers | 0.065 ± 0.122 | 0.028 | 0.001 | 0.474 |

Diet contributions were estimated from isotope mixing models created in MixSIAR. $\partial^{13}$C values were lipid normalized following the methods of McConnaughey and McRoy (1979). Mean diet contributions ± SD, with median, lower, and upper 95% Bayesian credible intervals are given.

## Discussion

Stable isotopes have proven useful in ecology for discerning migration patterns, feeding guilds, trophic levels, and seasonal diet shifts [21, 51–53]. In this study we investigated whether carbon and nitrogen stable isotopes could be used to track wild bee foraging in adjacent habitats within a reclaimed mine system. We expected bees to vary in isotope composition spatially and seasonally based on underlying differences in isotope fractionation of their food plants by habitat and month. We also sought to determine how taxon and social caste influence wild bee isotope signature, because there is no baseline data in the literature for North American species (but see [54] for $\partial^{15}$N values for bee amino acids). Bee isotope composition strongly reflected genus caste group (which was tied to month) and habitat. However, spatial and seasonal trends in bee isotope composition did not match those of their food plants. Flower $\partial^{13}$C was elevated in the field and edge compared to forest, and higher in late summer than in spring, as expected, but was confounded by other plant characteristics like growth form (woody versus herbaceous). Flower $\partial^{15}$N varied by plant taxon, but not predictably by habitat. Bee taxon and social group were more important predictors of a bee's isotope signature than either foraging habitat or month, likely because they are coupled with underlying differences in bee life history, diet, and timing of annual activity. Isotope mixing models were based on flower food sources that had high variation and overlap between habitats, so should be interpreted with caution. Models estimated similar habitat contributions to diets of spring *Andrena* and overwintered *Bombus* queens. Summer *Bombus* queens and workers were indistinguishable. Sweat bees (*Halictus*) were estimated to use comparatively more field flowers than others. Overall, isotope analysis offered more insight into each bee's diet than we would have gained from a one-time observation on one flower using traditional bee monitoring methods. However, the lack of difference in flower C and N isotope ratios by habitat is a major limitation of this technique in

this system. It is possible that in other less botanically diverse or better delineated habitats, the isotopic differences may be more pronounced.

## Trends in bee and flower $\partial^{13}$C

We expected bee $\partial^{13}$C to show the same trends in isotope composition by habitat as their food plants. They did not. $\partial^{13}$C trends of flowers followed our predictions based on presumed water stress: field flowers were enriched in $\partial^{13}$C relative to forest flowers, and edge flowers were intermediate between the two. Instead of mirroring the plants within their foraging habitat, bees netted in the field were depleted in $\partial^{13}$C compared to those captured along the forest edge. Similarly, social bees foraging in tropical pasture land were more depleted in $\partial^{13}$C values compared to those collected from nearby forest [11]. The mechanism for this pattern remains uncertain, but in our system, habitat is not independent of month and taxon, variables that also influenced $\partial^{13}$C values. Flower availability shifts from forest to field over the season, and bee taxa vary in the degree to which they overlap with the floral resources within each habitat.

Seasonal trends in isotope composition also did not align between bee abdomens and flowers. As expected if environmental factors strongly affected plant isotope fractionation, flowers were significantly enriched in $\partial^{13}$C in July when growing conditions were hot and dry. However, bees were enriched in $\partial^{13}$C in spring compared to late summer. In spring in Northeastern North America unpredictable weather can limit bees' foraging time and most bees have little nutritional reserves. Spring *Bombus* queens and *Andrena* typically overwinter as adults, and have likely already used most of their carbon-rich fat stores. Furthermore, cool, wet, weather and short foraging days may exacerbate their nutritional stress. Near starvation conditions may result in elevated $\partial^{13}$C. Indeed, in our study bees collected in April and May, especially overwintered spring *Bombus* queens and *Andrena carlini*, were enriched in $\partial^{13}$C compared to others. Taki et al. [55] reported a similar seasonal trend in $\partial^{13}$C of bees, with higher $\partial^{13}$C values in June honey bees compared to those from later months. Longer foraging days, drier weather, and high flower abundance in summer should reduce nectar carbon limitation and result in the lower $\partial^{13}$C values like those we observed in summer bees.

## Trends in bee and flower $\partial^{15}$N

We presumed field flowers would be more N-limited [56, 57] and, thus, incorporate more of the heavy nitrogen than forest flowers, and that $^{15}$N would bioaccumulate in bees accordingly. This pattern was found for arboreal ants in young second growth forests that were enriched in $\partial^{15}$N compared to ants from nearby mature forests, suggesting nitrogen limitation in earlier successional habitats [58]. In our study, field flowers were not enriched in $\partial^{15}$N compared to forest, suggesting that N was not as limiting in the field as expected. However, plants varied widely within habitat in their $\partial^{15}$N ratios. Bees netted from the field were less enriched in $\partial^{15}$N than those foraging along the edge. Brosi et al. [11] similarly found species specific differences in bee $\partial^{15}$N, but no overall enrichment in $\partial^{15}$N of pasture bees compared to forest. Plants in agricultural lands that have been treated with N-rich fertilizer should have lower N-limitation and lower $\partial^{15}$N values. In our study site, chemical fertilizer was applied to the field during the initial site reclamation approximately 15 years prior to the study, but not replenished thereafter, so should not have affected long-term N availability. Aside from grasses, the dominant plants in the field were herbaceous Fabaceae planted during reclamation (*Lotus corniculatus*, *Trifolium* spp., *Melilotus* spp.) [59], which associate with nitrogen-fixing rhizobia. Excess N exported to the soil by these legumes can alleviate N-limitation for other forbs [60, 61], and reduce $\partial^{15}$N.

We did not detect an increase in plant $\partial^{15}$N across the summer, despite expectations that volatilization of ammonia from warm dry soils would lead to seasonal increase in $^{15}$N concentration in the soil [27]. One possible explanation for our observation is rapid depletion of soil nitrogen in spring when water availability is high and plants and soil microbes grow quickly, which could result in increased $\partial^{15}$N of flower tissue in the spring. However, the nitrogen cycle is complex and $\partial^{15}$N in plant tissues is affected by a suite of factors beyond habitat and water availability, including soil fungal and bacterial pathways [62] and differences in physiology of individual species [63]. This complexity limits the generality of natural abundance $\partial^{15}$N isotope data.

Bee abdomens were enriched in $\partial^{15}$N in June—August relative to April and May, suggesting stronger N limitation in summer versus spring bees. This seasonal pattern was likely driven by the predominance of bumble bees in summer samples (which had elevated $\partial^{15}$N compared to other genera), but bee reproductive status may also have contributed. Bees have little muscle in their abdomens, so reproductive tissue is the primary nitrogen reservoir. Maturation of ovaries and eggs requires much nitrogen, which could lead to $\partial^{15}$N enrichment in reproductive individuals. Indeed, *Bombus* queens were the most enriched in $\partial^{15}$N of any bee group. However, solitary *Andrena* females were also reproducing in April and May, but had much lower $\partial^{15}$N values than *Bombus* queens. Summer *Bombus* workers and *Halictus* females were most likely both non-reproductive workers, without the high N requirements of egg laying, yet nonetheless *Bombus* had higher $\partial^{15}$N than *Halictus*. Therefore, taxon appears to have contributed more strongly to seasonal patterns in bee isotope composition than reproductive status. One possible mechanism for how taxon could have influenced $\partial^{15}$N is through brood provisioning behavior. A bee's primary source of nitrogen is the pollen it eats as a larva provisioned by its mother; adults fuel their metabolism with nectar. *Andrena* females and *Halictus* queens lay one egg per pollen ball then seal the cell, but *Bombus* queens lay a clutch of eggs that hatch and feed on a communal pollen mass [64]. Placement in the nest also determines how much *Bombus* larvae are fed; larvae near the center are fed more and turn into larger adults [65]. Food competition among larvae in *Bombus* may result in higher $\partial^{15}$N values compared to the other two genera.

## Use of isotope mixing models

Isotope mixing models had high source variation, and so were not fully able to source bees diets by habitat and should be interpreted with caution. Nonetheless, mixing models offered some insight. Models estimated similar habitat contributions to diets of spring *Andrena* and overwintered *Bombus* queens, but attributed less of their diet to forest resources than expected given the strong evidence in the literature that they rely on forest understory flowering herbs and shrubs [66–69]. *Bombus* queens and *Andrena* more closely resembled field flower $\partial^{13}$C because they were elevated in $\partial^{13}$C instead of depleted like forest flowers. Summer *Bombus* queens and workers were indistinguishable, with a high proportion attributed to edge flowers (likely because of their broad diets and because of their elevated $\partial^{15}$N similar to edge flowers sampled). Sweat bees (*Halictus*) were estimated to use comparatively more field flowers than others, consistent with what is known about their foraging. However, even spring bees were predicted to obtain more than half of their nutrition from field flowers. This may be a failure of the model to resolve habitat diet sources, or it could actually be the case as the field offered nectar-rich spring forage in the form of invasive flowering shrubs like *Elaeagnus umbellata* and *Lonicera morrowii*.

Bees use of field flowers is easily verified through observation, but bee foraging on forest resources is difficult to quantify. In this study, we spent > 6 hours observing primarily bee-

pollinated wildflowers in the forest understory but saw no bees visiting them, though low springtime visitation rates may be typical [67]. A patient observer can document visits to forest understory flowers, but verifying bee foraging on tree flowers requires extra determination and special equipment [66, 68, 69]. If isotope mixing models can reliably attribute a proportion of bees' diets to forest flowers, it would be a useful tool for inferring bees' use of forest plants, where bee monitoring is difficult and often incomplete. However, the lack of separation by habitat of our flowers, especially in $\partial^{15}N$, limits the interpretations that can be made from our mixing model results. Increased flower sampling from all habitats could improve the fit of isotope mixing models by reducing variation and overlap in potential food source isotope values. Many studies have reported carbon and nitrogen isotope natural abundances in roots, shoots, and leaves [70, 71], or fruit alone [72] but little is known about how much fractionation happens from vegetative to reproductive tissues [but see 73–75]. Honey $\partial^{13}C$ values from published studies [*e.g.*, 13] are not interchangeable with raw flower nectar because honey is highly concentrated. A baseline of flower C and N isotope values from a variety of habitats and times would help refine mixing model estimates. Since flower petals do not contribute to bee nutrition, removing the flower petals and using only fresh anthers with pollen and the base of the flower with nectaries may have better represented bee diets.

## Future directions

A deeper understanding of how much variation exists between individual bees and flowers and across taxa may allow for more reliable inference of wild bee habitat use from isotope composition. In addition, laboratory-based controlled feeding studies would provide insight into how sensitive a bee's isotope composition is to recent adult diet versus larval diet [76, 77]. Even with controlled laboratory studies laying the groundwork, species-specific variation in natural abundance of stable isotopes may limit the utility of C and N isotope analysis for revealing community-level trends in bee foraging. In our study, there were large differences between species of *Halictus* and *Andrena* in $\partial^{15}N$, although there were not noticeable differences in $\partial^{15}N$ among *Bombus* species. Pooling samples of individuals within species can help reduce within-taxa variation [21]. We pooled flowers from three individuals per species because we anticipated high variation between plants due to micro-scale differences in soil nutrients, but treated each bee individually, because bees feed from a large number of flowering plants.

To answer questions of bee habitat use, we agree with the conclusion of Brosi et al. [11] that isotope analysis would be best applied in agricultural systems, where chemical fertilizers create differences in N signatures between plants in agricultural versus non-agricultural areas to provide clearer habitat boundaries. Taki et al. [55] further support that idea with their finding that honey bee $\partial^{13}C$ and $\partial^{15}N$ values were significantly influenced by the proportion of forest and agricultural lands. Isotope enrichment studies, in which organisms are "labeled," or grown with an excess of a heavy isotope ($^{13}C$ via heavy $CO_2$ or $^{15}N$ via chemical fertilizer for plants or proteinaceous food source for insects), could be used to identify pollinators of plants that are of agricultural interest. For example, isotope labeled crop plants could be used to reveal the importance of native bees to crop pollination or be used to track bee dispersal between fields for specialists like squash bees (*Peponapis pruinosa*). Isotope analysis could also be used in combination with identification of nest pollen to confirm the importance of native mason bees *Osmia* for orchard fruit tree pollination because their nests are easily manipulated [78].

## Supporting information

**S1 Table. Bee and flower C and N stable isotope raw data from this study.**
(CSV)

**S1 File. This document includes the results of a preliminary laboratory feeding trial (text), analyses of bee head+thorax isotope composition (text, S5 and S6 Tables, S1-S3 Figs), and tables that further support the main manuscript results (S2-S4 Tables).**
(DOCX)

## Acknowledgments

Robert Burkhart and Max Frankenberry conducted pilot studies in the Goodell lab on bee stable isotope biology. Sophia Feller, Eda Wu, and Steven Novotny assisted with field and lab work. Wildlife biologist Lynda Andrews at Wayne National Forest allowed us access to the study site. Members of Dr. Andréa Grottoli's lab (OSU Earth Sciences) and Dr. Jim Bauer's lab (OSU, Evolution, Ecology, and Organismal Biology) provided advice on experimental design and equipment for isotope sample preparation.

## Author Contributions

**Conceptualization:** Jessie Lanterman Novotny, Karen Goodell.

**Data curation:** Jessie Lanterman Novotny.

**Formal analysis:** Jessie Lanterman Novotny.

**Funding acquisition:** Jessie Lanterman Novotny.

**Investigation:** Jessie Lanterman Novotny.

**Methodology:** Jessie Lanterman Novotny, Karen Goodell.

**Project administration:** Jessie Lanterman Novotny, Karen Goodell.

**Writing – original draft:** Jessie Lanterman Novotny.

**Writing – review & editing:** Jessie Lanterman Novotny, Karen Goodell.

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
