## [Decision Letter · Decision Letter 0]

10 Feb 2022

PONE-D-21-26062Utility of carbon and nitrogen stable isotopes for inferring wild bee (Hymenoptera: Apoidea) use of adjacent foraging habitatsPLOS ONE

Dear Dr. Novotny,

Thank you for submitting your manuscript to PLOS ONE. After careful consideration, we feel that it has merit but does not fully meet PLOS ONE’s publication criteria as it currently stands. Therefore, we invite you to submit a revised version of the manuscript that addresses the points raised during the review process.

Two experts in the field have provided critical and constructive comments with which I agree.

We look forward to receiving your revised manuscript.

Kind regards,

Renee M. Borges

Academic Editor

PLOS ONE

Journal Requirements:

4. Please upload a new copy of Figures 2 and 4 as the detail is not clear. Please follow the link for more information: " ext-link-type="uri" xlink:type="simple">https://blogs.plos.org/plos/2019/06/looking-good-tips-for-creating-your-plos-figures-graphics/"
" ext-link-type="uri" xlink:type="simple">https://blogs.plos.org/plos/2019/06/looking-good-tips-for-creating-your-plos-figures-graphics/"

5. We note that Figure 1 in your submission contain map images which may be copyrighted. All PLOS content is published under the Creative Commons Attribution License (CC BY 4.0), which means that the manuscript, images, and Supporting Information files will be freely available online, and any third party is permitted to access, download, copy, distribute, and use these materials in any way, even commercially, with proper attribution. For these reasons, we cannot publish previously copyrighted maps or satellite images created using proprietary data, such as Google software (Google Maps, Street View, and Earth). For more information, see our copyright guidelines: http://journals.plos.org/plosone/s/licenses-and-copyright.

Additional Editor Comments (if provided):

Two experts in the field have provided critical and constructive comments with which I agree.

Reviewers' comments:

Reviewer's Responses to Questions

**Comments to the Author**

1. Is the manuscript technically sound, and do the data support the conclusions?

Reviewer #1: Partly

Reviewer #2: Yes

2. Has the statistical analysis been performed appropriately and rigorously? 

Reviewer #1: Yes

Reviewer #2: Yes

3. Have the authors made all data underlying the findings in their manuscript fully available?

Reviewer #1: No

Reviewer #2: Yes

4. Is the manuscript presented in an intelligible fashion and written in standard English?

Reviewer #1: Yes

Reviewer #2: Yes

5. Review Comments to the Author

Reviewer #1: In this manuscript, Novotny and Goodell explored the utility of carbon and nitrogen stable isotopes for inferring bee use of different habitats. In doing so, they found that 13C and 15n varied with habitat, time, genus, and social caste but that bee isotopic values did not reflect spatial or temporal trends in the available food plants. This is an interesting result and does highlight the complexity of using stable isotopes for understanding bee diets. In fact, it is true what the authors say that many people do not use stable isotopes for bees, but I do have some major and minor comments listed below.

First, I think the biggest concern I have in this study is the lack of bees collected in forest habitat (Table S1). It doesn’t feel like a true test of habitat/resource use with zero observations, and I think this really impacts the validity of the mixing models later in the paper. Do I think bees could be using the floral resources in the woods…yes. But with no observational data confirming this then this paper really is a test of habitat use between field and edge.

Second, there is a lot of discussion about differential isotopic fractionation which is good to see. This is much too often left out of stable isotope papers. However, the arguments about water and nitrogen availability are a bit confusing as I couldn’t find a location in this manuscript where either water or nitrogen were measured (at least in my readings and word searches). In line 158-159, it is mentioned that habitats have “environmental differences that were likely to translate into differences in isotope fractionation of plants”, but this is not enough. This is problematic as I think you want to show these things are changing across the months if this is indeed the underlying mechanisms for differences in isotope values.

Third, the corrections for isotopic values seems quite odd (lines265-276). Why are these being done and is there a source that can be provided for justification of this method? Mainly, I am trying to understand why having more carbon in a sample should result in a sample being corrected.

Lines 81-83: This sentence feels oddly placed for this part of the introduction.

Line 115: Suggest changing “emerge” to “emerging” or remove “are” beforehand.

Line 137-139: Not sure how I feel about this result at the end of the introduction. This result appears out of no where and I think should likely be removed. I enjoyed the end of the intro finishing with predictions.

Lines 207-208: Do flower petals represent bee forage?

Lines 215-216: Abdomens could also contain actual food particles from recently digested food lowering their isotopic composition. In other arthropod studies, there is a lot of caution away from using abdomens for this reason. Also on lines 328-331, it is mentioned that abdomen was a better predictor but were the models actually different? This likely should be shown.

Line 231: Why were prepared samples put in a freezer after drying?

Line 315-316: Unfortunately I don’t agree. Additional sampling events should have taken place to confirm whether bees were in the forest or not. Simply saying they are despite not observing them there is problematic. (see major comment above)

Reviewer #2: The article by Novotny and Goodell investigates the utility of stable isotopes C and N to identify floral feeding patches of wild bees. Across three habitats (field, edge, and forest) the researchers found consistent isotope differences among bee species and across seasons, but these did not necessarily line up with isotope patterns in representative flowers from each habitat. The authors conclude that stable isotopes may have utility in comparing foraging across habitats that are more isotopically distinct (e.g., fertilized agricultural vs. wild habitats) and recommend future studies.

In general, I found the methodology to be rigorous and up to the standards for publication in PLOS ONE. I do have several points and questions that could be addressed to could help improve the manuscript:

Lines 107-109 - It would be helpful to discuss better how to take development into account. Can you reliably identify when foods would have been foraged that support the larval development of bees of different species? This seems like a major factor that could drive a mismatch between isotopic values of current flower blooms with cuticle content. I understand this may be difficult to estimate, but it would be worth discussing this in the introduction and explaining how past authors have dealt with this concern.

Lines 132-136 – I appreciate the inclusion of predictions, but I think they would be improved if they could be more specific. For prediction 1, do you expect specific habitats to be more water stressed than others? If so, you should be able to make a directional prediction about which habitats will have elevated N15 and C13 and when. I think it could also be worthwhile to break prediction two into separate points. For example, it seems that the main prediction is missing, which is that bees captured foraging in certain habitats will share isotopic signatures of the flowers blooming in those habitats. In addition, you have several predictions quickly mentioned in the preceding paragraph (lines 117-127) that I think would be relevant predictions to test (differences in solitary vs. social species, or differences in the variation isotope values associated with body size and foraging distance).

Lines 189-195 – This section contains results of the present study and would be better included in the results. I think the manuscript would benefit by including an initial section to the results that discusses spatial and seasonal patterns in bee and flower diversity. I also would like to see Table S1 added to the main text, and maybe even Table S2 if space is permitted. This would help readers like me, who are unfamiliar with bee communities in southeastern Ohio, to get a better sense of what the community looks like.

Lines 216-218 – It would be worth going into more detail about why abdomen content was used rather than head/thorax. In ants, almost every study explicitly excludes the abdomen (or gaster) because the storage of recent food items in the crop and relative differences in fat reserves among individuals can introduce bias into isotopic comparisons (see publications below). The comparisons between head/thorax and abdomen samples presented in the supplement help make the case for why abdomens were used for the current study, but it would be useful to add a paragraph to the discussion explain how lipids in the sample could have affected the results.

Tillberg C, McCarthy D, Dolezal A, Suarez A (2006) Measuring the trophic ecology of ants using stable isotopes. Insectes Sociaux 53, 65 – 69.

Post DM, Layman CA, Arrington DA, Takimoto G, Quattrochi J, Montana CG (2007) Getting to the fat of the matter: models, methods and assumptions for dealing with lipids in stable isotope analyses. Oecologia 152, 179 – 189.

Lines 382-383 – Looking at month across all bees would seem to confound the differences you see among species. To me, it isn’t clear that you are detecting any seasonal difference here besides the fact that different bee species are active at different times of the year. How could you account for this?

Minor comments:

Line 59 – Remove the word “and” (typo)

Line 75-77 – Since there are so few studies using stable isotopes to investigate the diet of bees, it would be helpful to cite work here using isotopes to investigate whether honey bees use human food sources in urban habitats:

Penick, C. A., Crofton, C. A., Holden Appler, R., Frank, S. D., Dunn, R. R., Tarpy, D. R. (2016). The contribution of human foods to honey bee diets in a mid-sized metropolis. Journal of Urban Ecology, 2(1).

Lines 137-139 – These sentences should go into the discussion, as they mention results of the present study.

Lines 158-159 – It would help to add 2-3 sentences explaining the environmental differences among habitats in more detail.

Line 202-203 – This sentence seems like it has a typo, as I could not fully understand the meaning. Did it mean to say that “We did NOT quantify forest flowers…”?

Line 277 – delete “and” before “genus” and add a comma.

Figure 3 – It would be nice to add a legend on the graphs to explain what the colors mean (rather than having to read each explanation in the figure caption).

Lines 462-466 – It seems that the patterns in bee foraging location are opposite of what you would predict (edge-collected bees were foraging mostly in fields, while field-collected bees were foraging evenly between field and forest). If so, it would be helpful to explain that these results do not align with your initial predictions.

Lines 492-501 – I thought this section was well written and certainly clarified the main point of the paper. Nice work.

6. PLOS authors have the option to publish the peer review history of their article (what does this mean?). If published, this will include your full peer review and any attached files.

Reviewer #1: No

Reviewer #2: No

---

## [Author Response · Author response to Decision Letter 0]

17 May 2022

Responses to Editors’ and Reviewers’ Comments

Manuscript PONE-D-21-26062

Academic Editor: Renee M Borges

Comments from the Editor

https://journals.plos.org/plosone/s/file?id=wjVg/PLOSOne_formatting_sample_main_body.pdf Edited manuscript body formatting according to style requirements. 

Edited title page according to style requirements.

2. In your Methods section, please provide additional information regarding the permits you obtained for the work. Please ensure you have included the full name of the authority that approved the field site access and, if no permits were required, a brief statement explaining why. Added how permission was obtained to access the study site.

Please update my data availability statement to read: All relevant data are within the manuscript and its Supporting Information files. Rather than make the raw data available through a third party repository, I will provide the raw data file in the Supporting Information of this paper through PLOS.

4. Please upload a new copy of Figures 2 and 4 as the detail is not clear. Please follow the link for more information: https://blogs.plos.org/plos/2019/06/looking-good-tips-for-creating-your-plos-figures-graphics/" For assistance preparing figures, please contact figures@plos.org. Figures have been revised to improve image quality using the PACE platform.

5. We note that Figure 1 in your submission contain map images which may be copyrighted. All PLOS content is published under the Creative Commons Attribution License (CC BY 4.0), which means that the manuscript, images, and Supporting Information files will be freely available online, and any third party is permitted to access, download, copy, distribute, and use these materials in any way, even commercially, with proper attribution. For these reasons, we cannot publish previously copyrighted maps or satellite images created using proprietary data, such as Google software (Google Maps, Street View, and Earth). For more information, see our copyright guidelines: http://journals.plos.org/plosone/s/licenses-and-copyright. We require you to either (1) present written permission from the copyright holder to publish these figures specifically under the CC BY 4.0 license, or (2) remove the figures from your submission. Please upload the completed Content Permission Form or other proof of granted permissions as an "Other" file with your submission. 

I requested permission from the publishers of the National Land Cover Data (NLCD) 2011 at the US Geological Survey to publish my study site map (Fig 1). They replied that no permission was needed to publish my map because the NLCD map is public domain, and is not copyrighted. The publishing manager for the NLCD, Jon Dewitz, signed the request to publish form anyway in case you need it.

Comments from Reviewer #1

In this manuscript, Novotny and Goodell explored the utility of carbon and nitrogen stable isotopes for inferring bee use of different habitats. In doing so, they found that 13C and 15n varied with habitat, time, genus, and social caste but that bee isotopic values did not reflect spatial or temporal trends in the available food plants. This is an interesting result and does highlight the complexity of using stable isotopes for understanding bee diets. In fact, it is true what the authors say that many people do not use stable isotopes for bees, but I do have some major and minor comments listed below.

First, I think the biggest concern I have in this study is the lack of bees collected in forest habitat (Table S1). It doesn’t feel like a true test of habitat/resource use with zero observations, and I think this really impacts the validity of the mixing models later in the paper. Do I think bees could be using the floral resources in the woods…yes. But with no observational data confirming this then this paper really is a test of habitat use between field and edge. 

We have carefully considered the reviewer’s point, but have decided not to remove forest from our mixing model analysis. One of our main motivations for doing this study was to cultivate alternate methods (beyond the traditional netting and pan trapping) for assessing bee resource use because of the difficulty and expense of observing bees foraging in forests and areas of particularly dense vegetation (Introduction, line 62-64 in original manuscript before revisions). We hoped the isotope composition might be a reliable predictor of bee forest use that could be used to supplement traditional monitoring methods. There is strong evidence from the literature and from the authors’ personal observations that spring bees use forest understory flowers in our study region. Forest comprised 90% of the land use in the area surrounding the reclaimed field, so all of our focal bee taxa would have had access to forest resources within their flight ranges. 

Second, there is a lot of discussion about differential isotopic fractionation which is good to see. This is much too often left out of stable isotope papers. However, the arguments about water and nitrogen availability are a bit confusing as I couldn’t find a location in this manuscript where either water or nitrogen were measured (at least in my readings and word searches). In line 158-159, it is mentioned that habitats have “environmental differences that were likely to translate into differences in isotope fractionation of plants”, but this is not enough. This is problematic as I think you want to show these things are changing across the months if this is indeed the underlying mechanisms for differences in isotope values.

The focus of this study was to look for patterns in bee and flower isotope values between habitats and taxa because that is largely absent from the literature. Differences between habitats and across the season in soil N and water availability have been previously established and are summarized in the Introduction (lines 99 – 106 in the original manuscript). In response to the Reviewer’s concern, we have now revised the Introduction and Methods so that we do not unintentionally imply that we measured soil water and N in this study. We did obtain weather data on average precipitation, relative humidity, temperature, and wind speed, from the airport nearest to the research site for the two weeks preceding each sampling date. However, weather variables were not related to flower isotope signature and did not improve the amount of variation that models explained (likely because average precipitation is low on/around days that are optimal for bee field work), so this analysis was not included in the Methods or in the final analyses. 

Third, the corrections for isotopic values seems quite odd (lines265-276). Why are these being done and is there a source that can be provided for justification of this method? Mainly, I am trying to understand why having more carbon in a sample should result in a sample being corrected.

Before natural abundance of isotope values can be compared, it is common practice to verify that there are not underlying patterns in the data that could confound the effects of environmental variables. In response to Reviewer 1’s concern here and also Reviewer 2’s question below about lipid content of the samples, we have now applied the more commonly used McConnaughey and McRoy (1979) lipid normalization method to our bee ∂13C values instead of our previous C correction method. This required us to re-run analyses and remake figure panels for all analyses were bee ∂13C was used. For consistency, we also updated the analyses and figures in the supplementary information to lipid normalize head+thorax ∂13C values as well. The Methods Data Analysis section was revised accordingly and references added. These changes did not change the major patterns in the data or our conclusions.

Note: Post et al. (2007) concluded that the lipid normalization method does not seem to work for plants like it does with animal tissues, so flower isotope values were not adjusted.

Lines 81-83: This sentence feels oddly placed for this part of the introduction. Revised to better transition from the previous sentence.

Line 115: Suggest changing “emerge” to “emerging” or remove “are” beforehand. Removed “are” beforehand.

Line 137-139: Not sure how I feel about this result at the end of the introduction. This result appears out of no where and I think should likely be removed. I enjoyed the end of the intro finishing with predictions. We misinterpreted the requirements of the journal, and this section has been removed. In the instructors for submission, authors are instructed to: “Conclude with a brief statement of the overall aim of the work and a comment about whether that aim was achieved (https://journals.plos.org/plosone/s/submission-guidelines).” However, I agree with Reviewers 1 and 2 that our results summary at lines 137 – 139 is not needed.

Lines 207-208: Do flower petals represent bee forage? Petals do not contribute to bee nutrition. However, not a lot is known about isotope fractionation in the reproductive parts of the plant, so I used whole flowers to get an integrated measure of flower isotope composition. Fresh anthers with pollen and the base of the flower (where the nectaries are) minus the petals may have better represented bee diets. I added a sentence about this to the Discussion in the paragraph on how to improve the fit of mixing models.

Lines 215-216: Abdomens could also contain actual food particles from recently digested food lowering their isotopic composition. In other arthropod studies, there is a lot of caution away from using abdomens for this reason. Also on lines 328-331, it is mentioned that abdomen was a better predictor but were the models actually different? This likely should be shown.

In response to the reviewer’s major comment above, we revised the method of correcting for lipids in the abdomen samples, which can disproportionately lower the ∂13C. We also revised text in the Data Analysis section of the Methods where we explained how our field methods helped account for recent food in the digestive tract. The result that the reviewer is referring to at line 328 was from linear discriminant analysis rather than multiple linear models. There are no R2 values in that type of analysis because it does not generate typical model output, but rather uses ∂13C values to predict group membership of the samples in some category of interest like habitat or genuscaste. For models of how habitat and genuscaste influenced isotope composition, model R2adj values are given in the model summary tables in the Results for abdomen models and in the Supporting Information for head+thorax models. 

Line 231: Why were prepared samples put in a freezer after drying? Placing the samples in the freezer was not necessary, but was done out of an abundance of caution and did not alter the samples.

Line 315-316: Unfortunately I don’t agree. Additional sampling events should have taken place to confirm whether bees were in the forest or not. Simply saying they are despite not observing them there is problematic. (see major comment above). We added a sentence to this section of the Methods to further justify our inclusion of forest in isotope mixing models: “Also, one of the motivations of this study was to explore the use of isotope analysis for supplementing traditional ways of monitoring bee habitat use, particularly in habitats where observation of foraging bees is restricted by equipment, vegetation, and weather.” We sampled in the forest during the April and May surveys (peak time for forest understory flowers and bees), but did not observe any bees there even with 6 hours of observation on clear weather days (lines 193-195 in original manuscript). In the summer, on each sampling date we did a preliminary 20 minute walk through the forest but there were no flowers from which to net so no further timed surveys were conducted.

Comments from Reviewer #2

The article by Novotny and Goodell investigates the utility of stable isotopes C and N to identify floral feeding patches of wild bees. Across three habitats (field, edge, and forest) the researchers found consistent isotope differences among bee species and across seasons, but these did not necessarily line up with isotope patterns in representative flowers from each habitat. The authors conclude that stable isotopes may have utility in comparing foraging across habitats that are more isotopically distinct (e.g., fertilized agricultural vs. wild habitats) and recommend future studies.

In general, I found the methodology to be rigorous and up to the standards for publication in PLOS ONE. I do have several points and questions that could be addressed to could help improve the manuscript:

Lines 107-109 - It would be helpful to discuss better how to take development into account. Can you reliably identify when foods would have been foraged that support the larval development of bees of different species? This seems like a major factor that could drive a mismatch between isotopic values of current flower blooms with cuticle content. I understand this may be difficult to estimate, but it would be worth discussing this in the introduction and explaining how past authors have dealt with this concern. The reviewer raises an excellent point that the differences in resources fed to larvae (from one month to one year before we collected that adult) versus nectar eaten recently as an adult likely contribute to the mismatch between flower and bee isotope values. The effect of larval resources on adult isotope composition (whether it is likely to raise or lower ∂13C and ∂15N) is not consistent across the taxa we used, because of differences in timing and duration of annual activity between genera. Halictus and Bombus larval food should have been from the same season but one month before the adult (so isotopically lighter if water was less limiting when the larvae were feeding in spring/early summer). But Andrena larval food would have been from the previous spring, and we do not know the extent to which isotope signature of plants varies between years. This issue was an important reason why we analyzed the abdomen data instead of the head+thorax data, even though other insect isotope studies have avoided abdomens. However, we feel it would interrupt the flow of the main argument to explain it in the Introduction. Instead, we addressed this on lines 218 – 221 of the Methods. There no precedent in the literature for how to account for this issue in using stable isotopes for bee foraging ecology. Brosi et al. (2009) is the only other study that we are aware of that has used bee isotopes to distinguish feeding habitats, and they did not account for this issue, other than treating each bee as an independent sample in models due to variability in larval provisions. Controlled laboratory feeding trials are needed to determine the contribution of larval versus adult diets to adult bee tissues.

Lines 132-136 – I appreciate the inclusion of predictions, but I think they would be improved if they could be more specific. For prediction 1, do you expect specific habitats to be more water stressed than others? If so, you should be able to make a directional prediction about which habitats will have elevated N15 and C13 and when. I think it could also be worthwhile to break prediction two into separate points. For example, it seems that the main prediction is missing, which is that bees captured foraging in certain habitats will share isotopic signatures of the flowers blooming in those habitats. In addition, you have several predictions quickly mentioned in the preceding paragraph (lines 117-127) that I think would be relevant predictions to test (differences in solitary vs. social species, or differences in the variation isotope values associated with body size and foraging distance).

The predictions were revised per Reviewer 1’s suggestions. Prediction 2 was broken into two parts, and prediction 1 was revised to be more specific / directional. This study was designed to try to detect spatial foraging patterns, while taking into the account of time of the season. We would have needed to use a greater variety of bees to test predictions on sociality or body size, and are not able to do that with this dataset. Because we did not know the level of background variation we would encounter across taxa, we choose to test only three common representative genera instead of a broad spectrum of the bee diversity available.

Lines 189-195 – This section contains results of the present study and would be better included in the results. I think the manuscript would benefit by including an initial section to the results that discusses spatial and seasonal patterns in bee and flower diversity. I also would like to see Table S1 added to the main text, and maybe even Table S2 if space is permitted. This would help readers like me, who are unfamiliar with bee communities in southeastern Ohio, to get a better sense of what the community looks like.

As the Reviewer requested, the summary of how many bees were collected was moved from the Methods to the beginning of the Results and Tables S1 and S2 were moved from the Supporting Information to the main manuscript. We are not able to use the data from this study to present overall spatial and seasonal patterns of bee or flower diversity. For the purpose of this study, we only sampled a small select subset of bee and flower communities at the site. 

Lines 216-218 – It would be worth going into more detail about why abdomen content was used rather than head/thorax. In ants, almost every study explicitly excludes the abdomen (or gaster) because the storage of recent food items in the crop and relative differences in fat reserves among individuals can introduce bias into isotopic comparisons (see publications below). The comparisons between head/thorax and abdomen samples presented in the supplement help make the case for why abdomens were used for the current study, but it would be useful to add a paragraph to the discussion explain how lipids in the sample could have affected the results.

Tillberg C, McCarthy D, Dolezal A, Suarez A (2006) Measuring the trophic ecology of ants using stable isotopes. Insectes Sociaux 53, 65 – 69.

Post DM, Layman CA, Arrington DA, Takimoto G, Quattrochi J, Montana CG (2007) Getting to the fat of the matter: models, methods and assumptions for dealing with lipids in stable isotope analyses. Oecologia 152, 179 – 189.

Lipid content is of concern when measuring ∂13C of animal tissues. Some people pretreat the samples to remove lipids, but Post et al (2007) concluded that it was preferable to mathematically correct later. Therefore, rather than pretreat and risk losing important biological information bees’ nutritional status, we left abdomens intact as bulk tissue and applied a mathematical lipid correction. In response to Reviewer 1 and 2’s suggestions on how to better deal with lipids, we have now changed our correction method to the McConnaughey and McRoy (1979) lipid normalization equation. After lipid normalization, our abdomen ∂13C values were enriched instead of depleted compared to head+thorax ∂13Clipid normalized. We feel that after the mathematical correction for the lipid content of the sample was applied, the results were not influenced by lipid content enough to warrant a new paragraph in the Discussion.

Tillberg et al 2006 found ant worker abdomens were depleted in ∂13C compared to head+thorax (likely because fat reserves in abdomens or recent food in the crop can bring down their ∂13C). Rather than apply a mathematical lipid correction to their abdomen values, instead they used the head+thorax to avoid the issue. For our study, we used the abdomens (after accounting for lipids) because we found evidence in the literature that the abdomen would better reflect recent adult diet and more closely match flower isotope composition. To help deal with the issue of recent food bringing down ∂13C, in the field our bees were left in the vials for a minimum of two hours before freezing and they typically defecated and regurgitated nectar from the crop during that time before they were freezer killed. 

Lines 382-383 – Looking at month across all bees would seem to confound the differences you see among species. To me, it isn’t clear that you are detecting any seasonal difference here besides the fact that different bee species are active at different times of the year. How could you account for this? To account for the fact that month is confounded with genus, we have revised the season trends section to use only a subset of data from one genuscaste group that was represented across the season – bumble bee workers. 

Minor comments:

Line 59 – Remove the word “and” (typo). Removed “and”

Line 75-77 – Since there are so few studies using stable isotopes to investigate the diet of bees, it would be helpful to cite work here using isotopes to investigate whether honey bees use human food sources in urban habitats: Penick, C. A., Crofton, C. A., Holden Appler, R., Frank, S. D., Dunn, R. R., Tarpy, D. R. (2016). The contribution of human foods to honey bee diets in a mid-sized metropolis. Journal of Urban Ecology, 2(1). Added suggested reference and revised sentence accordingly.

Lines 137-139 – These sentences should go into the discussion, as they mention results of the present study. We misinterpreted the requirements of the journal, and this section has been removed. In the instructors for submission, authors are instructed to: “Conclude with a brief statement of the overall aim of the work and a comment about whether that aim was achieved (https://journals.plos.org/plosone/s/submission-guidelines).” However, I agree with Reviewers 1 and 2 that our results summary at lines 137 – 139 is not needed.

Lines 158-159 – It would help to add 2-3 sentences explaining the environmental differences among habitats in more detail. Revised to: “Based on the literature, we also assumed that forest and field had differences in water and nitrogen availability that were likely to translate into differences in isotope fractionation of the plants [23 – 27].” Further details of habitats classification were already in the Methods at lines 182-184.

Line 202-203 – This sentence seems like it has a typo, as I could not fully understand the meaning. Did it mean to say that “We did NOT quantify forest flowers…”? We did mean to say “not quantified.” Revised.

Line 277 – delete “and” before “genus” and add a comma. Revised as suggested.

Figure 3 – It would be nice to add a legend on the graphs to explain what the colors mean (rather than having to read each explanation in the figure caption). We tried to add a legend as the reviewer requested but there was not space to place a separate consistently-formatted legend in the same position on each panel.

Lines 462-466 – It seems that the patterns in bee foraging location are opposite of what you would predict (edge-collected bees were foraging mostly in fields, while field-collected bees were foraging evenly between field and forest). If so, it would be helpful to explain that these results do not align with your initial predictions. After re-analysis described above, we removed this subsection of the results because the habitat groupings were less informative than the genuscaste bee grouping factor in mixing models. 

Lines 492-501 – I thought this section was well written and certainly clarified the main point of the paper. Nice work. We thank the reviewer for this positive feedback.

---

## [Decision Letter · Decision Letter 1]

24 Jun 2022

Utility of carbon and nitrogen stable isotopes for inferring wild bee (Hymenoptera: Apoidea) use of adjacent foraging habitats

PONE-D-21-26062R1

Dear Dr. Novotny,

We’re pleased to inform you that your manuscript has been judged scientifically suitable for publication and will be formally accepted for publication once it meets all outstanding technical requirements.

Kind regards,

Renee M. Borges

Academic Editor

PLOS ONE

Additional Editor Comments (optional):

Reviewers' comments:

Reviewer's Responses to Questions

**Comments to the Author**

1. If the authors have adequately addressed your comments raised in a previous round of review and you feel that this manuscript is now acceptable for publication, you may indicate that here to bypass the “Comments to the Author” section, enter your conflict of interest statement in the “Confidential to Editor” section, and submit your "Accept" recommendation.

Reviewer #1: All comments have been addressed

Reviewer #2: All comments have been addressed

2. Is the manuscript technically sound, and do the data support the conclusions?

Reviewer #1: Yes

Reviewer #2: Yes

3. Has the statistical analysis been performed appropriately and rigorously? 

Reviewer #1: Yes

Reviewer #2: Yes

4. Have the authors made all data underlying the findings in their manuscript fully available?

Reviewer #1: Yes

Reviewer #2: Yes

5. Is the manuscript presented in an intelligible fashion and written in standard English?

Reviewer #1: Yes

Reviewer #2: Yes

6. Review Comments to the Author

Reviewer #1: This is my second review of this manuscript and I have primarily focused on the original review responses. Overall, the authors have responded well to the first set of revisions, but I still wonder about the bee collections in forest. I agree with the authors that having alternative methods for understanding how bees use resources is important. Stable isotopes could be one of those methods that helps us uncover unknown interactions. Yet if forest truly made up 90% of the land type in the surrounding area and sampling occurred during peak flowering time (lines 197-198) then surely some bees should have been collected. I understand the difficulty in collecting in such habitats, but there is a missing gap in data that could be vital for understanding if bees are present at this particular site using resources in this particular habitat (i.e. forests). Perhaps the information in table 5 highlights why there might not have been many bees in the forest with the mixing models suggesting low contributions from forest plants. I have thought about this particular discussion point for a while, but I do not want to cause any further delays and thus leave the final decision up to the editor.

Minor comments:

On correcting isotope values with lipids– I still don’t think that these modifications/corrections are necessary for isotope values in this study. This seems like more work than is necessary especially since things like flowers were homogenized with petals that do not contribute to bee nutrition (and thus may change the isotope values of the food items). Not trying to be picky here, but just stating that I think this is more work than is necessary.

Lines 267-268: What units are lipid content in? mg?

Reviewer #2: I was happy to see how all comments were addressed. Interpreting stable isotope data can be difficult, and this study certainly showcases the nuances required to understand such comparisons. Nice work.

7. PLOS authors have the option to publish the peer review history of their article (what does this mean?). If published, this will include your full peer review and any attached files.

Reviewer #1: No

Reviewer #2: No

---

## [Editor Report · Acceptance letter]

1 Jul 2022

PONE-D-21-26062R1 

Utility of carbon and nitrogen stable isotopes for inferring wild bee (Hymenoptera: Apoidea) use of adjacent foraging habitats 

Dear Dr. Novotny:

I'm pleased to inform you that your manuscript has been deemed suitable for publication in PLOS ONE. Congratulations! Your manuscript is now with our production department. 

Kind regards, 

on behalf of

Professor Renee M. Borges 

Academic Editor

PLOS ONE